# *Ureaplasma* Species Modulate Cytokine and Chemokine Responses in Human Brain Microvascular Endothelial Cells

**DOI:** 10.3390/ijms20143583

**Published:** 2019-07-22

**Authors:** Christine Silwedel, Christian P. Speer, Axel Haarmann, Markus Fehrholz, Heike Claus, Nicolas Schlegel, Kirsten Glaser

**Affiliations:** 1University Children’s Hospital, University of Wuerzburg, Josef-Schneider-Str. 2, 97080 Wuerzburg, Germany; 2Department of Neurology, University of Wuerzburg, Josef-Schneider-Str. 11, 97080 Wuerzburg, Germany; 3Institute for Hygiene and Microbiology, University of Wuerzburg, Josef-Schneider-Str. 2, 97080 Wuerzburg, Germany; 4Department of Surgery I, University of Wuerzburg, Oberduerrbacherstr. 6, 97080 Wuerzburg, Germany

**Keywords:** *Ureaplasma urealyticum*, *Ureaplasma parvum*, neuroinflammation, meningitis, blood–brain barrier, HBMEC

## Abstract

*Ureaplasma* species are common colonizers of the adult genitourinary tract and often considered as low-virulence commensals. Intraamniotic *Ureaplasma* infections, however, facilitate chorioamnionitis and preterm birth, and cases of *Ureaplasma*-induced neonatal sepsis, pneumonia, and meningitis raise a growing awareness of their clinical relevance. In vitro studies are scarce but demonstrate distinct *Ureaplasma*-driven impacts on immune mechanisms. The current study addressed cytokine and chemokine responses upon exposure of native or lipopolysaccharide (LPS) co-stimulated human brain microvascular endothelial cells (HBMEC) to *Ureaplasma urealyticum* or *U. parvum*, using qRT-PCR, RNA sequencing, multi-analyte immunoassay, and flow cytometry. *Ureaplasma* exposure in native HBMEC reduced monocyte chemoattractant protein (MCP)-3 mRNA expression (*p* < 0.01, vs. broth). In co-stimulated HBMEC, *Ureaplasma* spp. attenuated LPS-evoked mRNA responses for C-X-C chemokine ligand 5, MCP-1, and MCP-3 (*p* < 0.05, vs. LPS) and mitigated LPS-driven interleukin (IL)-1α protein secretion, as well as IL-8 mRNA and protein responses (*p* < 0.05). Furthermore, *Ureaplasma* isolates increased C-X-C chemokine receptor 4 mRNA levels in native and LPS co-stimulated HBMEC (*p* < 0.05). The presented results may imply immunomodulatory capacities of *Ureaplasma* spp. which may ultimately promote chronic colonization and long-term neuroinflammation.

## 1. Introduction

*Ureaplasma urealyticum* and *U. parvum* represent some of the smallest self-replicating pathogens and are generally considered as commensals, given their common colonization of the adult genitourinary tract [1]. Nonetheless, maternal–fetal transmission seems to occur frequently, and intraamniotic *Ureaplasma* infection may significantly contribute to chorioamnionitis and preterm birth [2,3,4]. The clinical relevance of a postnatal *Ureaplasma* infection, however, is still the subject of discussion and controversy [5]. *Ureaplasma* species were found responsible for invasive infections in immunocompromised adults and may furthermore cause pneumonia, sepsis, and meningitis in preterm and term neonates, with highest risk at low gestational ages [5,6,7,8,9,10,11,12]. Neonatal meningitis and neuroinflammation may provoke profound brain injury, possibly having long-term health implications [13,14,15]. Intraventricular hemorrhage (IVH) and white matter disease, both relevant neurological morbidities of prematurity, have been associated with inflammation [16,17,18]. Some studies even described an association between IVH and *Ureaplasma* colonization in preterm infants [19,20]. Considering these severe sequelae as well as reported *Ureaplasma* detection rates of up to 20% in the cord blood and/or cerebrospinal fluid (CSF) of preterm infants, it is likely that *Ureaplasma* spp. have to be regarded as being of considerable clinical relevance [2,5,20]. Nonetheless, clinical and in vitro data addressing *Ureaplasma*-driven neuroinflammation and underlying pathomechanisms are limited.

Inflammation is a condition carefully balanced by numerous mediators. These include pro-inflammatory cytokines such as tumor necrosis factor (TNF)-α, interleukin (IL)-1α, IL-1β, and IL-6; the anti-inflammatory cytokines IL-1 receptor antagonist (RA) and IL-10; the chemokine IL-8; monocyte chemoattractant proteins (MCP); and C-X-C chemokine ligands (CXCL) and receptors (CXCR) [21]. Matrix metalloproteinases (MMP) contribute to chemotaxis and degrade membranes [22]. Within the brain, inflammatory mediators are expressed primarily by residential macrophages such as microglia, but also by astrocytes and oligodendrocytes, as well as vascular endothelial cells [21]. Particularly noteworthy in neuroinflammation is the blood–brain barrier (BBB), which is responsible for maintaining the homeostasis of the central nervous system (CNS) and regulates the entry of macromolecules, microbes, and immune cells into the brain [23,24]. Compromised integrity of the BBB is common to many neuroinflammatory morbidities [23,24].

We recently established a cell culture model of *Ureaplasma* meningitis, using human brain microvascular endothelial cells (HBMEC) as integral components of the BBB. For the first time, we could provide in vitro evidence of *Ureaplasma*-induced apoptosis and barrier impairment in HBMEC [25,26]. Subsequently, the current study assessed *Ureaplasma*-driven cytokine and chemokine responses in native and, to mimic co-infections, *Escherichia coli* lipopolysaccharide (LPS)-primed HBMEC. To account for regulation of immune responses at the levels of transcription and translation, we correlated *Ureaplasma*-induced mRNA and protein expression for relevant inflammatory mediators using quantitative real-time reverse transcriptase polymerase chain reaction (qRT-PCR), RNA sequencing, multi-analyte immunoassay, and flow cytometry (Appendix A).

## 2. Results

### 2.1. Pro-inflammatory Cytokine and Chemokine Responses in HBMEC upon Stimulation with Ureaplasma Isolates or LPS

*Ureaplasma* stimulation of HBMEC for 30 h resulted in decreased MCP-3 mRNA expression (qRT-PCR: *U. urealyticum* serovar 8 (Uu8) 0.34-fold ± 0.30, *p* = 0.0381; *U. parvum* serovar 3 (Up3) 0.21-fold ± 0.13, *p* = 0.0008, vs. control), an effect that remained significant when compared to a broth control (Figure 1d). The broth itself moderately induced MCP-3 (Figure 1d,e). *Ureaplasma*-driven MCP-3 mRNA results were not reflected by altered protein secretion when analyzed at 24 and 48 h (Figure 1f).

Stimulation of native HBMEC with *Ureaplasma* isolates did not evoke significant mRNA or protein responses for IL-1α, MCP-1, CXCL5, or IL-8 (Figure 1), and furthermore, *Ureaplasma* stimulation did not have any impact on mRNA or protein expression of IL-1β, IL-6, or TNF-α (data not shown). MMP-9 mRNA and protein were not expressed in native HBMEC, nor did we observe any *Ureaplasma*-induced increases (data not shown).

LPS, on the other hand, induced distinct mRNA and protein responses for most given mediators (Figure 1). Compared to control cells, LPS-stimulated HBMEC showed higher IL-1α mRNA (*p* < 0.01) and protein (*p* < 0.001) levels (Figure 1a–c), elevated MCP-3 mRNA (*p* < 0.01) and protein (*p* < 0.001) expression (Figure 1d–f), increased MCP-1 mRNA (*p* < 0.001) and protein (*p* < 0.05) levels (Figure 1g–i), a higher abundance of CXCL5 mRNA (*p* < 0.05, vs. control) and protein (*p* < 0.01) (Figure 1j–l), and increased IL-8 mRNA and protein expression (*p* < 0.01; Figure 1m–p). IL-1β mRNA, IL-6 mRNA, and secreted protein, as well as TNF-α mRNA and secreted protein, were also significantly increased by LPS stimulation of HBMEC (data not shown). MMP-9 mRNA and protein were not inducible by LPS (data not shown).

### 2.2. Pro-inflammatory Cytokine and Chemokine Responses in Co-stimulated HBMEC

Whereas *Ureaplasma* spp. did not evoke distinct responses for most cytokines and chemokines in native HBMEC (Figure 1), we observed a significant impact on cells co-stimulated with LPS. Both *Ureaplasma* isolates significantly down-regulated LPS-induced IL-1α protein secretion after 48 h (Uu8+LPS 0.26-fold ± 0.11, *p* = 0.0169, vs. LPS; Uu8+LPS 0.21-fold ± 0.09, *p* = 0.0076, vs. broth+LPS; Up3+LPS 0.22-fold ± 0.10, *p* = 0.0122, vs. LPS; Up3+LPS 0.18-fold ± 0.08, *p* = 0.0080, vs. broth+LPS; Figure 2c), although we did not observe distinct *Ureaplasma*-driven mRNA effects in co-stimulated HBMEC (Figure 2a,b). *Ureaplasma* spp. attenuated MCP-3 mRNA responses in LPS-stimulated HBMEC (qRT-PCR: Uu8+LPS 0.08-fold ± 0.06, *p* = 0.0251, vs. LPS; Uu8+LPS 0.04-fold ± 0.03, *p* = 0.0299, vs. broth+LPS; Up3+LPS 0.17-fold ± 0.08, *p* = 0.0131, vs. LPS; Up3+LPS 0.09-fold ± 0.04, *p* = 0.0292, vs. broth+LPS; Figure 2d,e). The abundance of secreted MCP-3 protein was also reduced in co-stimulated cells; however, it did reach statistical significance (Figure 2f). Furthermore, LPS-evoked MCP-1 mRNA levels were mitigated upon *Ureaplasma* exposure (qRT-PCR: Uu8+LPS 0.35-fold ± 0.08, *p* = 0.0397, vs. LPS; Uu8+LPS 0.28-fold ± 0.07, *p* = 0.0181, vs. broth+LPS; Up3+LPS 0.48-fold ± 0.3, *p* = 0.0132, vs. LPS; Up3+LPS 0.40-fold ± 0.25, *p* = 0.0002, vs. broth+LPS; Figure 2g,h). Broth effects were mainly responsible for an increase in MCP-1 protein secretion in co-stimulated cells (Figure 2i). Both *Ureaplasma* isolates significantly mitigated LPS-induced CXCL5 mRNA responses after 30 h (qRT-PCR: Uu8+LPS 0.30-fold ± 0.10, *p* = 0.0309, vs. LPS; Uu8+LPS 0.29-fold ± 0.10, *p* = 0.0174, vs. broth+LPS; Up3+LPS 0.35-fold ± 0.11, *p* = 0.0274, vs. LPS; Up3+LPS 0.34-fold ± 0.11, *p* = 0.0167, vs. broth+LPS; Figure 2j,k). Co-stimulated HBMEC also showed a trend towards reduced CXCL5 protein secretion (Figure 2l). LPS-induced IL-8 mRNA levels were reduced upon 30 h of Uu8 exposure in HBMEC (qRT-PCR: Uu8+LPS 0.49-fold ± 0.20, *p* = 0.0276, vs. LPS; Uu8+LPS 0.37-fold ± 0.15, *p* = 0.0059, vs. broth+LPS; Figure 2m,n). Although the abundance of secreted IL-8 protein was not influenced (Figure 2o), both *Ureaplasma* isolates significantly down-regulated LPS-driven intracellular IL-8 protein responses after 24 h (Uu8+LPS 0.26-fold ± 0.16, *p* = 0.0068, vs. LPS; Uu8+LPS 0.20-fold ± 0.12, *p* < 0.0001, vs. broth+LPS; Up3+LPS 0.42-fold ± 0.20, *p* = 0.0434, vs. LPS; Up3+LPS 0.32-fold ± 0.15, *p* = 0.0012, vs. broth+LPS; Figure 2p) and 48 h (Uu8+LPS 0.08-fold ± 0.21, *p* = 0.0130, vs. LPS; Uu8+LPS 0.04-fold ± 0.11, *p* < 0.0001, vs. broth+LPS; Up3+LPS 0.01-fold ± 0.03, *p* = 0.0255, vs. LPS; Up3+LPS 0.006-fold ± 0.02, *p* = 0.0001, vs. broth+LPS; Figure 2p). *Ureaplasma* spp. did not affect LPS-evoked effects regarding IL-1β, IL-6, TNF-α, or MMP-9 (data not shown).

### 2.3. Anti-inflammatory Cytokine Responses in Mono- and Co-stimulated HBMEC

Anti-inflammatory IL-1RA and IL-10 showed neither relevant basal expression in HBMEC nor induction on mRNA or protein levels upon exposure to *Ureaplasma* or LPS or by co-stimulation with *Ureaplasma* isolates and LPS (data not shown).

### 2.4. CXCR4 Responses in Mono- and Co-stimulated HBMEC

We observed significantly increased CXCR4 mRNA levels in HBMEC exposed to *Ureaplasma* isolates for 30 h (qRT-PCR: Uu8 3.3-fold ± 0.8, *p* = 0.0144; Up3 3.1-fold ± 0.6, *p* = 0.0083, vs. broth; Figure 3a,b). LPS stimulation similarly enhanced CXCR4 mRNA expression in HBMEC (*p* < 0.01, vs. control; Figure 3a,b). In HBMEC co-stimulated with LPS, Up3 significantly enhanced CXCR4 mRNA levels after a 30 h stimulation period (RNA-seq: Up3+LPS 2.3-fold ± 0.2, *p* = 0.0322, vs. LPS; 2.7-fold ± 0.2, *p* = 0.0092, vs. broth+LPS; Figure 3d).

## 3. Discussion

Inflammation is a fastidiously balanced condition orchestrated by cytokines, chemokines, and their respective receptors [27]. Although a number of case reports suggest a profound clinical relevance of *Ureaplasma* spp. in neonatal neuroinflammation [11,12], we are only beginning to unravel the underlying pathomechanisms. For the first time, this study provides insights into meticulous *Ureaplasma*-driven cytokine and chemokine responses in HBMEC. Our results demonstrate that differential modulations of immune defense mechanisms rather than fierce direct pro-inflammation initiate *Ureaplasma*-induced neuroinflammation.

Whereas LPS, widely used to mimic bacterial infections in vitro, evoked distinct pro-inflammatory responses in HBMEC, expressed by mRNA and protein increases for CXCL5, IL-1α, IL-1β, IL-6, IL-8, MCP-1, MCP-3, and TNF-α, *Ureaplasma* spp. failed to induce similar effects (Figure 1). MCP-3 mRNA levels were even down-regulated upon 30 h of *Ureaplasma* stimulation (Figure 1d).

Chemokines such as CXCL5, IL-8, MCP-1, and MCP-3 are essential for leukocyte recruitment, and cytokines like IL-1α, IL-1β, IL-6, and TNF-α function as intracellular messengers promoting inflammatory responses, with IL-1α particularly initiating cytokine responses and neutrophil influx to sites of inflammation [28,29]. An increase of these pro-inflammatory mediators is a physiological phenomenon in infections and necessary for a sufficient immune defense. A lack thereof, as seen in *Ureaplasma*-stimulated cells, may impede pathogen elimination and facilitate chronic *Ureaplasma* infections. Interestingly, our in vitro results are underlined by a clinical study demonstrating that *Ureaplasma* detection within the CSF of preterm infants of <1500 g birth weight was not associated with increased CSF IL-1β, IL-6, and TNF-α levels [20].

*Ureaplasma* spp. not only failed to evoke pro-inflammatory cytokine and chemokine responses in native HBMEC but furthermore modulated LPS-evoked immune reactions in co-stimulated cells (Table 1). We observed distinct *Ureaplasma*-driven attenuation of LPS-induced CXCL5, IL-1α, IL-8, MCP-1, and MCP-3 responses (Figure 2). These findings in co-stimulated cells might reflect deficient immune responses to secondary pathogens in vivo. *Ureaplasma* spp. may thus increase the susceptibility of the CNS to additional infections. Underlining our in vitro findings, animal models demonstrated mitigation of LPS-induced immune responses in chronic *Ureaplasma*-colonized fetal sheep [30], and a higher risk for neonatal sepsis was described in *Ureaplasma*-colonized preterm infants [31].

Furthermore, chemokines are no longer considered only as pro-inflammatory mediators but are also known to exert protective and regenerative effects in the CNS [24]. MCP-1, for example, is necessary for recruitment of monocytes and macrophages into the CNS not only in inflammation but also under physiological conditions [24]. MCP-1 is expressed in human fetal brains [32], and up-regulated MCP-1 was found to increase the tolerance for brain ischemia in mice [33]. MCP-1 may thus have a neuroprotective function. Down-regulation of LPS-induced MCP-1 and other chemokine responses by *Ureaplasma* spp., as observed in this study (Figure 2), may therefore indicate decreased resilience of the CNS against secondary inflammatory or non-infectious impacts in vivo, facilitating subsequent brain injury. This may not only explain the potential association of *Ureaplasma* spp. with IVH [19,20]. Given, for example, the influence of ischemia on the development of white matter disease in preterm infants [18], *Ureaplasma*-driven reduced CNS resilience may causally contribute to several other neurological morbidities of prematurity.

Our results moreover demonstrate a relevant increase of CXCR4 mRNA expression in HBMEC upon *Ureaplasma* exposure, the extent of which is comparable to effects evoked by LPS (Figure 3a,b). CXCR4 is a G-protein-coupled seven-transmembrane receptor expressed by B and T cells, monocytes, and endothelial cells [34,35]. It mediates inflammatory cell migration and contributes to physiological neurodevelopmental processes, but it was furthermore found to be increased in viral encephalitis and autoimmune neuroinflammation [35,36,37]. Accordingly, CXCR4 antagonists were shown to protect the BBB and reduce inflammatory responses within the CNS [38]. Whereas the exact interactions between CXCR4 and the BBB are yet to be determined, previous studies suggest a negative impact of CXCR4 on tight junction (TJ) proteins responsible for maintaining barrier function [38]. The CXCR4 mRNA increase in HBMEC exposed to *Ureaplasma* spp. may therefore be an indicator of *Ureaplasma*-driven impairment of TJ proteins via CXCR4 engagement, ultimately resulting in compromised BBB integrity. This may be even more pronounced in the event of co-infections, as indicated by further amplified CXCR4 mRNA responses in HBMEC co-stimulated with *Ureaplasma* spp. and LPS (Figure 3c,d, Table 1).

We recently demonstrated atypical chemokine receptor 3 induction in HBMEC exposed to *Ureaplasma* spp. [26]. This receptor, formerly known as CXCR7, may also facilitate BBB breakdown [26]. Moreover, *Ureaplasma*-induced apoptosis in HBMEC may further impair barrier properties [25]. We previously revealed reduced adhesion properties in *Ureaplasma*-exposed HBMEC [25]. BBB disruption therefore appears to be of particular relevance in *Ureaplasma*-driven neuroinflammation. The BBB usually provides both an anatomical barrier and a cytokine-dependent immunological control to regulate passage into the CNS [24]. Compromised integrity of the BBB is considered a key factor in neuroinflammation [23,39]. BBB impairment by *Ureaplasma* spp. may ultimately allow inflammatory cell influx into the CNS but also facilitate pathogen entry into the brain. The resulting consequence may be long-term intracerebral inflammation. This is particularly relevant when considering not only cases describing chronic *Ureaplasma* meningitis but also those studies suggesting an association of *Ureaplasma* spp. with IVH and cerebral palsy in preterm neonates, both conditions being promoted by inflammation [5,11,12,19,20,40].

Taking together the exaggerated CXCR4 responses in co-stimulated cells and the *Ureaplasma*-driven mitigation of LPS-induced cytokine and chemokine responses, *Ureaplasma* spp. appear to exert an immunomodulatory capacity in the event of co-infection. This may be even more pronounced in the event of non-simultaneous inflammatory hits [30] as occurring, for example, in chorioamnionitis. The augmentation of some effects evoked by secondary impact factors and the attenuation of others may both impair important CNS immune defense mechanisms just in those cells intended to protect the brain. This immunomodulation may represent a key feature of the initiation of *Ureaplasma*-driven neuroinflammation, ultimately facilitating invasive infection and consecutive tissue inflammation even in the absence of “direct” *Ureaplasma*-evoked pro-inflammatory cytokine and chemokine effects. Considering the frequency of co-infections and polymicrobial colonization in preterm infants, this finding may be of particular relevance.

We did not detect any relevant basal or stimulation-induced anti-inflammatory IL-10 or IL-1RA responses in HBMEC. This is in line with previous studies describing no LPS-induced IL-10 production in brain endothelial cells [41]. Interestingly, IL-10 is considered protective against apoptosis by some authors [42], and the lack thereof in HBMEC may indicate a particular vulnerability to apoptosis, underlining our previous results [25] and ultimately again impairing BBB integrity.

The mechanisms behind *Ureaplasma*-driven immune modulation are not yet fully understood. However, *Ureaplasma* spp. are known to employ Toll-like receptor (TLR) signaling [43,44]. Given that TLR activate signaling pathways which induce numerous immune mediators, including inflammatory cytokines [45], TLR engagement may be responsible for *Ureaplasma*-induced cytokine and chemokine effects.

The *Ureaplasma*-driven effects observed in this study became obvious only after longer stimulation periods (Figure 1, Figure 2 and Figure 3), whereas LPS evoked earlier responses (Figure 1). This may be in line with often subtle, subclinical *Ureaplasma* infections, which become relevant over the course of time rather than as an acute infection. Case reports describe chronic *Ureaplasma* meningitis persisting for several months [11,12].

This is the first study addressing *Ureaplasma*-driven cytokine and chemokine responses in HBMEC. The study is strengthened by the use of two viable *Ureaplasma* isolates. Although subtle differences between Uu8 and Up3 were apparent regarding IL-8 and CXCR4 mRNA responses in co-stimulated HBMEC (Figure 2m,n, Figure 3c,d), the isolates did not evoke distinctly diverging effects. Another strength is the assessment of cytokine and chemokine responses at the level of mRNA and protein expression.

In vitro studies can only partially resemble in vivo conditions, where external influencing factors and complex interactions have to be taken into account. Furthermore, given the particular vulnerability of preterm and term neonates, the use of an adult cell line may be considered a potential limitation of this study. Therefore, future studies ought to be based on neonatal rather than adult cells and should comprise tissue models or co-culture systems more closely resembling the complex in vivo BBB.

The findings presented in this study partially differ from results obtained previously, in which our group demonstrated *Ureaplasma*-induced pro-inflammatory cytokine and chemokine responses in primary human neonatal and adult monocytes but also a modulation of LPS-evoked effects [46,47]. *Ureaplasma* spp. therefore appear to exert tissue-specific pro- and anti-inflammatory effects, both potentially with severe negative ramifications. Either way, our consistent in vitro findings of *Ureaplasma*-driven immune modulation underline their pathogenicity and profound clinical relevance.

## 4. Materials and Methods

### 4.1. Bacterial Strains and Culture Conditions

Serovar 8 of *U. urealyticum* (Uu8) and serovar 3 of *U. parvum* (Up3) were acquired from the American Tissue Culture Collection (ATCC; Uu8: ATCC 27618, Up3: ATCC 27815). Isolates were propagated in a liquid in-house medium (“broth”) containing 82% autoclaved pleuropneumonia-like organism medium (Becton, Dickinson & Company, Franklin Lakes, NJ, USA), 10% heat-inactivated horse serum (*v*/*v*), 1% urea (*w*/*v*), and 0.002% phenol red (*w*/*v*) (all: Sigma-Aldrich, St. Louis, CA, USA). The medium was passed through a 0.2 µm filter membrane (Sartorius, Goettingen, Germany) and adjusted to pH 6.5. An endotoxin level of <0.06 EU/mL was verified (ToxinSensor™ Endotoxin Detection System, GenScript, Piscataway, NJ, USA).

As described previously [26], 10-fold serial dilutions of *Ureaplasma* cultures were incubated to obtain 1 × 10^9^–1 × 10^10^ color-changing units (CCU)/mL of viable organisms. The corresponding *Ureaplasma* DNA amounted to 5 × 10^7^–6 × 10^8^ copies/mL (Institute of Medical Microbiology and Hospital Hygiene, Duesseldorf, Germany). Bacterial viability was confirmed by simultaneous culture on selective agar plates (*Mycoplasma Ureaplasma* agar, medco Diagnostika GmbH, Ottobrunn, Germany).

### 4.2. Cell Line and Culture Conditions

Non-immortalized adult HBMEC (ACBRI 376, Cell Systems, Kirkland, WA, USA) were grown to confluence in gelatin (Serva Electrophoresis, Heidelberg, Germany) coated T-75 culture flasks (Greiner Bio-One, Frickenhausen, Germany) in a humid atmosphere at 37 °C with 5% CO_2_. Cells were propagated in RPMI-1640 medium (Sigma-Aldrich) supplemented with 10% fetal calf serum (Thermo Fisher Scientific, Waltham, MA, USA), 10% Nu-Serum (BD Biosciences, San Jose, CA, USA), 2 mM L-glutamine, 1 mM sodium pyruvate, 1% minimum essential medium with non-essential amino acids (all: Thermo Fisher), 5 U/mL heparin (Biochrom, Berlin, Germany), and 0.3% endothelial cell growth supplement (Cell Systems). As described previously [26], confluent monolayers were expanded, and recently thawed cells of passage 8 were used for all experiments. Previous experiments had confirmed basic endothelial characteristics [26].

### 4.3. Stimulation Assays

HBMEC were seeded in gelatin-coated 6-well culture plates (Greiner Bio-One) at a density of 2 × 10^5^ cells/well and grown to confluence for 48 h. Monolayers were washed, and 1 mL of fresh growth medium was added per well. *E. coli* serotype 055:B5 LPS (Sigma-Aldrich) was applied to a subgroup of HBMEC at a concentration of 100 ng/mL. As described previously [26], 250 µL broth harboring 1 × 10^9^–1 × 10^10^ CCU of *Ureaplasma* isolates was added per mL of HBMEC medium to native or LPS co-stimulated HBMEC. Cultures were incubated for 4 and 30 h for mRNA analysis, whereas incubation periods of 24 and 48 h were chosen for flow cytometry and multi-analyte immunoassay. The doses and durations of stimulation were determined in previous experiments [26]. To acknowledge potentially confounding broth effects, cells exposed to the broth control and unstimulated HBMEC served as negative controls.

### 4.4. RNA Extraction and Reverse Transcriptase PCR (RT-PCR)

Total RNA was extracted using a NucleoSpin^®^ RNA Kit (Macherey-Nagel, Dueren, Germany), eluted in 60 μL RNAse-free H_2_O (Macherey-Nagel), quantified using a Qubit^®^ 2.0 Fluorometer (Thermo Fisher), and stored at −80 °C until reverse transcription. RT-PCR was performed with 1 μg of total RNA using a High Capacity cDNA Reverse Transcription Kit (Thermo Fisher). First-strand cDNA was diluted 1:10 with nuclease-free H_2_O (Sigma-Aldrich) and stored at −20 °C until analysis.

### 4.5. Quantitative Real-Time RT-PCR (qRT-PCR)

For quantitative detection of mRNA, cDNA was analyzed in duplicates of 25 μL reaction mixture containing 12.5 μL iTaq™ Universal SYBR^®^ Green Supermix (Bio-Rad Laboratories, Hercules, CA, USA), 0.5 μL nuclease-free H_2_O, and 1 μL of a 10 μM solution of forward and reverse primers (Sigma-Aldrich) as indicated in Table 2. Analysis was performed using an Applied Biosystems^®^ 7500 Real-Time PCR System (Thermo Fisher) with a two-step PCR protocol including an initial denaturation at 95 °C for 10 min and 40 cycles of 95 °C for 15 s and 60 °C for 1 min. At the end of every run, melt curve analysis was used to verify single PCR products. Amplification was normalized to the reference gene hypoxanthine phosphoribosyltransferase (*HPRT*) *1* and mean fold changes in mRNA expression were calculated according to the ΔΔC_T_ method [48]. Experiments were repeated five times (*n* = 5).

### 4.6. RNA Sequencing

Total RNA was extracted using a NucleoSpin^®^ RNA Kit (Macherey-Nagel), and samples were stored at −80 °C until further processing. Experiments were repeated three times (*n* = 3). Libraries were prepared at the Core Unit Systems Medicine, University of Wuerzburg, Germany, using the Illumina TruSeq stranded mRNA Kit (Illumina, San Diego, CA, USA) with 700 ng of input RNA and 13 PCR cycles. Thirteen or fourteen libraries were pooled and sequenced on a NextSeq 500 (Illumina) with a read length of 75 nucleotides, producing about 34–40 million raw reads per library. The Illumina TruSeq adaptors were cleaved with the help of cutadapt (version 1.14) [49], and reads were trimmed keeping the quality drop value below a mean of Q20. FastQC 0.11.5 (Braham bioinformatics, Cambridge, UK) [50] was used for assessing the read quality, number of duplicates, and presence of adapter sequences. Processed sequences were mapped to the human genome by employing the short read aligner STAR (version 2.5.2b) [51] with genome and annotation files from GENCODE (version 25—March 2016 freeze, GRCh38). For all samples, the proportion of reads mapped to the human reference genome ranged between 76% and 90% in total. Sequences aligning to specific genes were quantified using the bedtools subcommand intersect (version 2.15.0) [52], and differentially expressed genes were identified with the help of DESeq2 (version 1.16.1) [53]. The significance threshold for Benjamini–Hochberg corrected *p*-values was set at 0.05. To compare different groups, reads per kilo base per million mapped reads (RPKM) were calculated for individual genes using DGEList and the RPKM function from edgeR [54].

### 4.7. Multi-Analyte Immunoassay

Following stimulation, supernatants were collected and stored at −80 °C until analysis. Using Luminex^®^ multiplex kits and xPonent^®^ software (Merck group, Darmstadt, Germany), concentrations of secreted proteins were determined with the help of a standard curve. The lower detection limits were 11.0 pg/mL (CXCL5), 0.6 pg/mL (IL-1α), 0,9 pg/mL (IL-1β), 6.3 pg/mL (IL-1RA), 2.5 pg/mL (IL-6), 1.8 pg/mL (IL-8), 1.8 pg/mL (IL-10), 2.2 pg/mL (MCP-1), 5.0 pg/mL (MCP-3), 16.7 pg/mL (MMP-9), and 0.49 pg/mL (TNF-α), and values underneath were set to 0. Samples were analyzed in duplicate, and experiments were repeated five times (*n* = 5).

### 4.8. Flow Cytometry

For Fc-receptor blocking, harvested cells were incubated with gamunex (Grifols, Frankfurt, Germany) at a concentration of 4 mg/mL. Cells were separated by centrifugation and stained with a Brilliant Violet conjugated antibody to cluster of differentiation (CD) 31 (BV 510, BD Biosciences), a PerCP conjugated antibody to MMP-9 (Assay Pro, St. Charles, MO, USA), and Fixable Viability Dye eFluor^TM^ 780 (eBioScience, Thermo Fisher), a dye labelling dead cells. After centrifugation, cells were resuspended in phosphate-buffered saline (PBS, Sigma-Aldrich) containing 1% human serum (HS, Biochrom GmbH), fixed using fixation buffer (BioLegend, San Diego, CA, USA), and permeabilized in permeabilization wash buffer (BioLegend). Cells were then stained with antibodies to IL-8 (Alexa Fluor 488 conjugated, BioLegend) and IL-10 (APC conjugated, BioLegend). After centrifugation and resuspension in PBS/HS, samples were read on a FACSCanto™ II flow cytometer (BD Biosciences). A minimum of 10,000 events was acquired to be analyzed with FACSDiva v6.1.3 software (BD Biosciences). A side scatter height versus forward scatter width dot plot was used to exclude doublets, and cells were gated for viable, CD31-positive events. Experiments were repeated at least five times (*n* ≥ 5).

### 4.9. Statistical Analysis

Results were analyzed with the help of Prism^®^ 6 software (GraphPad Software, San Diego, CA, USA), using a one-way ANOVA with Tukey’s multiple comparisons test. Results with a corrected *p*-value of <0.05 were considered as significantly differentially expressed. Data are shown as means ± standard deviation (SD).

## 5. Conclusions

In marked contrast to other pathogens, *Ureaplasma* spp. rarely cause fierce CNS infections but rather induce subtle, long-term neuroinflammation. For the first time, our results convey insights into underlying cytokine and chemokine coherences. This in vitro study does not provide evidence for “classic” *Ureaplasma*-driven pro-inflammation in HBMEC but rather suggests an immunomodulatory capacity of *Ureaplasma* spp. in co-infections in vivo. Particularly in the presence of a secondary impact factor, *Ureaplasma* spp. may (i) mitigate pro-inflammatory cytokine and chemokine responses, impeding effective immune defense and allowing pathogen persistence; (ii) hamper neuro-protective mechanisms, increasing CNS vulnerability; and (iii) compromise BBB integrity, allowing pathogen and inflammatory cell influx into the CNS. All of these effects may ultimately facilitate invasive infections, chronic colonization, long-term neuroinflammation, and subsequent brain injury. *Ureaplasma* spp. therefore have to be considered relevant pathogens in neonatal neuroinflammation, particularly in the event of co-infection.

## Figures and Tables

**Figure 1 ijms-20-03583-f001:**
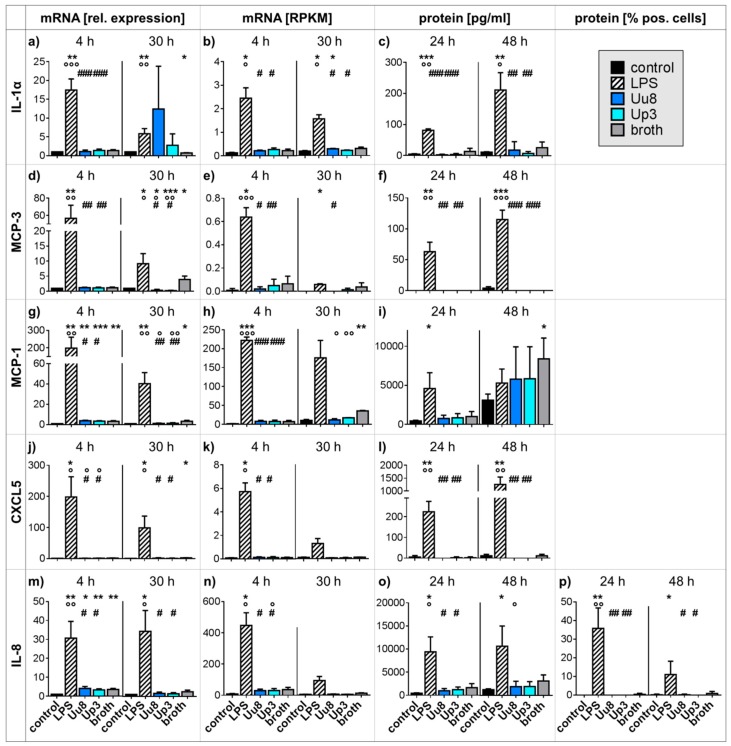
Cytokine and chemokine expression in HBMEC exposed to *Ureaplasma* spp. or LPS. qRT-PCR (**a**, **d**, **g**, **j**, **m**), RNA sequencing (**b**, **e**, **h**, **k**, **n**), and multi-analyte immunoassay (**c**, **f**, **i**, **l**, **o**) were applied for determination of quantitative mRNA expression and protein secretion. Intracellular IL-8 was additionally quantified by flow cytometry (**p**). Data are presented as means ± SD and were obtained from *n* ≥ 3 independent experiments (* *p* < 0.05, ** *p* < 0.01, *** *p* < 0.001 vs. unstimulated control; ° *p* < 0.05, °° *p* < 0.01, °°° *p* < 0.001 vs. broth; # *p* < 0.05, ## *p* < 0.01, ### *p* < 0.001 vs. LPS). CXCL: C-X-C chemokine ligand; HBMEC: human brain microvascular endothelial cells; IL: interleukin; LPS: lipopolysaccharide; MCP: monocyte chemoattractant protein; RPKM: reads per kilo base per million mapped reads; SD: standard deviation; Up3: *Ureaplasma parvum* serovar 3; Uu8: *Ureaplasma urealyticum* serovar 8.

**Figure 2 ijms-20-03583-f002:**
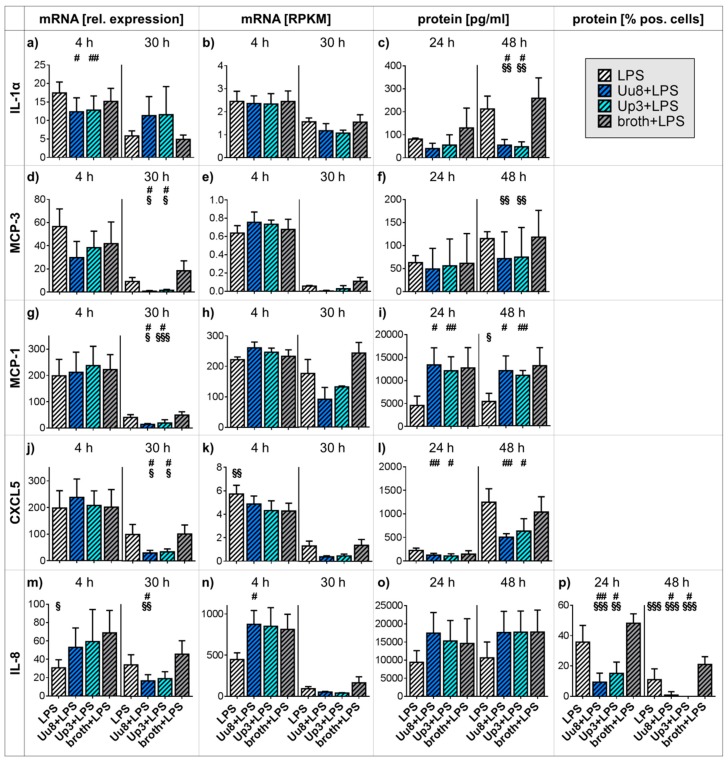
Cytokine and chemokine expression in HBMEC co-stimulated with *Ureaplasma* spp. and LPS. Expression of mRNA and protein secretion were determined using qRT-PCR (**a**, **d**, **g**, **j**, **m**), RNA sequencing (**b**, **e**, **h**, **k**, **n**), and multi-analyte immunoassay (**c**, **f**, **i**, **l**, **o**). Intracellular IL-8 was additionally quantified by flow cytometry (**p**). Data are presented as means ± SD and were obtained from *n* ≥ 3 individual experiments (# *p* < 0.05, ## *p* < 0.01 vs. LPS; § *p* < 0.05, §§ *p* < 0.01, §§§ *p* < 0.001 vs. broth+LPS). CXCL: C-X-C chemokine ligand; HBMEC: human brain microvascular endothelial cells; IL: interleukin; LPS: lipopolysaccharide; MCP: monocyte chemoattractant protein; RPKM: reads per kilo base per million mapped reads; SD: standard deviation; Up3: *Ureaplasma parvum* serovar 3; Uu8: *Ureaplasma urealyticum* serovar 8.

**Figure 3 ijms-20-03583-f003:**
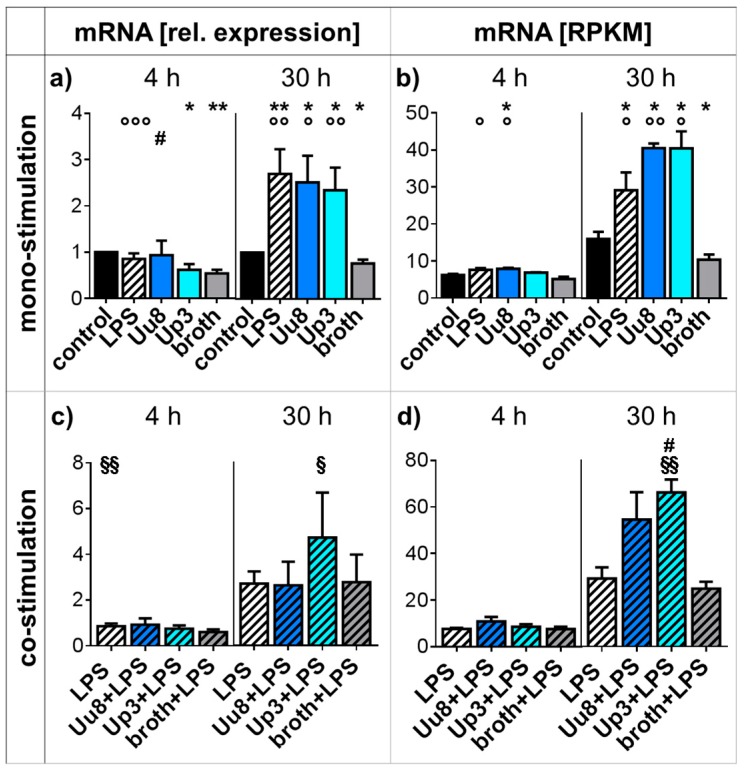
CXCR4 responses in mono- and co-stimulated HBMEC. CXCR4 mRNA levels in mono-stimulated cells were assessed by qRT-PCR (**a**) and RNA sequencing (**b**). In co-stimulated HBMEC, qRT-PCR (**c**) and RNA sequencing (**d**) were used to determine CXCR4 mRNA levels. Data are presented as means ± SD and were obtained from *n* ≥ 3 independent experiments (* *p* < 0.05, ** *p* < 0.01 vs. unstimulated control; ° *p* < 0.05, °° *p* < 0.01, °°° *p* < 0.001 vs. broth; # *p* < 0.05 vs. LPS; § *p* < 0.05, §§ *p* < 0.01 vs. broth+LPS). CXCR: C-X-C chemokine receptor; HBMEC: human brain microvascular endothelial cells; LPS: lipopolysaccharide; RPKM: reads per kilo base per million mapped reads; SD: standard deviation; Up3: *Ureaplasma parvum* serovar 3; Uu8: *Ureaplasma urealyticum* serovar 8.

**Table 1 ijms-20-03583-t001:** Effects of *Ureaplasma* spp. on LPS-evoked cytokine and chemokine responses and potential in vivo relevance. *Ureaplasma* spp. modulated LPS-induced cytokine and chemokine responses. Impacts were either antagonistic (LPS effects reduced) or additive (LPS effects enhanced). Significantly modulated readout parameters are listed for each mediator as well as the potential clinical relevance of modulated responses. BBB: blood–brain barrier; CXCL: C-X-C chemokine ligand; CXCR: C-X-C chemokine receptor; IL: interleukin; LPS: lipopolysaccharide; MCP: monocyte chemoattractant protein. ↑: increased.

	LPS Effect	LPS + *Ureaplasma* spp.	Potential In Vivo Relevance
**CXCL5**	mRNA and protein ↑	antagonistic (mRNA)	reduced immune responses
**CXCR4**	mRNA ↑	additive (mRNA)	compromised BBB integrity
**IL-1α**	mRNA and protein ↑	antagonistic (protein)	reduced immune responses
**IL-8**	mRNA and protein ↑	antagonistic (mRNA, protein)	reduced immune responses
**MCP-1**	mRNA and protein ↑	antagonistic (mRNA)	reduced immune responses
**MCP-3**	mRNA and protein ↑	antagonistic (mRNA)	reduced immune responses

**Table 2 ijms-20-03583-t002:** Primers used for qRT-PCR.

Name	Gene Symbol	Sequence Accession No.	Orientation	Sequence [5′ to 3′]
CXCL5	*CXCL5*	NM_002994.5	forward	GTTGCGTTTGTTTACAGACC
			reverse	TTTCCTTGTTTCCACCGT
CXCR4	*CXCR4*	NM_001008540.2	forward	AAGACCACAGTCATCCTC
			reverse	GTTCTCAAACTCACACCCT
HPRT1	*HPRT1*	NM_000194.2	forward	CTGGCGTCGTGATTAGTG
			reverse	AGTCCTGTCCATAATTAGTCC
IL-1α	*IL1α*	NM_000575.4	forward	ATGGATCAATCTGTGTCTCTG
			reverse	GCTTGATGATTTCTTCCTCTG
IL-1β	*IL1β*	NM_000576.2	forward	TTCATTGCTCAAGTGTCTG
			reverse	GCACTTCATCTGTTTAGGG
IL-8	*IL8*	NM_000584.4	forward	CAGTGCATAAAGACATACTCC
			reverse	TTTATGAATTCTCAGCCCTC
IL-10	*IL10*	NM_000572.3	forward	GCTGTCATCGATTTCTTCC
			reverse	GTCAAACTCACTCATGGCT
MCP-1	*CCL2*	NM_002982.4	forward	GCTGTGATCTTCAAGACC
			reverse	AAGTCTTCGGAGTTTGGG
MCP-3	*CCL7*	NM_006273.3	forward	CTGAGACCAAACCAGAAACC
			reverse	TATTAATCCCAACTGGCTGAG
MMP-9	*MMP9*	NM_004994.3	forward	GGATGGGAAGTACTGGCGA
			reverse	CTCCTCAAAGACCGAGTCC
TNF-α	*TNFα*	NM_000594.4	forward	CAGCCTCTTCTCCTTCCT
			reverse	GGGTTTGCTACAACATGG

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
