# Peer review of "Ureaplasma Species Modulate Cytokine and Chemokine Responses in Human Brain Microvascular Endothelial Cells"

_ijms, 2019, doi:10.3390/ijms20143583_

Round 1

Reviewer 1 Report

ijms-526080

Ureaplasma species mitigate immune defense strategies in human brain microvascular endothelial cells by modulation of cytokine and chemokine responses

The manuscript by Silwedel et al. examines the cytokine and chemokine responses of human brain microvascular endothelial cells to in vitro Ureaplasma spp. exposure. Ureaplasmas are associated with chorioamnionitis and neonatal morbidity and mortality, including neonatal meningitis and intraventricular haemorrhage. Despite this, there is a paucity of studies that have investigated the mechanisms of ureaplasma-induced neuroinflammation. Therefore, this manuscript addresses an important area of research.

Overall, the manuscript is well written but there are numerous areas in which the paper can be improved:

1.                   Title: the title is quite convoluted and could be simplified and more impactful if it was reworded to: “Ureaplasma species modulate cytokine and chemokine responses in human brain microvascular endothelial cells”.

2.                   Lines 37-42: The last three sentences within the abstract should be deleted as these are speculative and are not supported directly by data within the manuscript.

3.                   Line 47: It is not necessary to define that “U.” refers to “Ureaplasma” as this is common scientific nomenclature. At the first mention, the bacterial name should be written in full (e.g. Ureaplasma urealyticum) and all mentions after this are U. urealyticum.

4.                   Line 48: “low-virulent commensals” should be changed to “commensals” (i.e. delete low-virulent as this is already implied by the word commensal)

5.                   Line 52: It is not necessary to define “spp.” as this is a common abbreviation

6.                   Line 69: It should mentioned that resident macrophage populations within the brain (microglia and perivascular macrophages) are the primary cells that express immune mediators

7.                   Line 72: The phrase “shields the brain from external impact factors” is vague. This could be rewritten as “…responsible for maintaining homeostasis of the central nervous system and regulating the entry of macromolecules, microbes and immune cells into the brain”.

8.                   Line 78: What is the rationale for examining LPS-primed HBMECs? Please explain why this is important.

9.                   Line 82: This table is not useful and should not be referred to in the introduction. The reference to the table should either be moved to the results or materials and methods section, or deleted completely (this is my preference as I do not feel that the table is necessary)

10.                Line 87: In this results section, the authors have highlighted MCP-3 data first, which is shown in Figure 1G-I. When referring to the figure, it would make a lot more sense if the MCP-3 data was presented as Figure 1A-C and that reference to each of the figure panels should occur in chronological order in the main text. It is difficult to refer to the figure from the main text in the current format.

11.                Line 90: The authors highlight in the main text that fig 1G shows a decrease in MCP-3 mRNA expression compared to untreated controls via relative expression at 30 h. However, this result is not corroborated by the RNA sequencing (fig 1h) and protein secretion (fig 1i) data. The authors should address this explicitly and justify in the discussion why they believe that the mRNA relative expression data is valid given that it is not supported by other approaches. Two out of three approaches do not demonstrate statistically significant differences therefore are the mRNA relative expression data likely to be “real” or biologically relevant?

12.                Figures: All abbreviations used within figures should be defined within the figure legend. Please clarify the units for each of the y axes in figures 1-3

13.                Figure 2i: Please check the statistical analysis of the 48 h data as it seems based on the data presented and the size or the error bars that these data are unlikely to be statistically significant

14.                Figure 3c: Please check the statistical analysis of the 30 h data as it seems based on the data presented and the size of the error bars that these data are unlikely to be statistically significant

15.                Discussion: the authors should further discuss the limitations of their in vitro model. For example, it is stated that human brain microvascular endothelial cells are used to model the integral parts of the blood-brain barrier; however, it should be noted that HBMECs in monolayer cultures do not accurately model the blood-brain barrier, which is comprised of the ‘neurovascular unit’ consisting of brain microvascular endothelial cells, astrocytic foot processes, pericytes, neurons and microglia. Therefore, it should be mentioned that in order to study the effects of ureaplasmas on the blood-brain barrier in a physiologically relevant model, a co-culture system involving multiple cell types should be considered.

16.                Line 288: Please specify which “selective agar” was used.

17.                Line 300: Please comment whether HBMECs formed tight junctions/increased transendothelial electrical resistance in this study. If this was not performed/measured, please explain why as this is a key component of studying the blood-brain barrier.

18.                Line 308: Please explain why different time points were selected for flow cytometry and multi-analyte immunoassays compared to RNA studies.

19.                Line 305: How many technical replicates were included in each experiment (e.g duplicate/triplicate wells)?

20.                Line 312: How were samples prepared for RNA extraction? Were individual wells within an experiment pooled to obtain enough starting material to get sufficient RNA yields?

21.                Line 327: It is stated that experiments were repeated 5 times. Are the data shown in the figures pooled from the 5 experiments, or is data from one representative experiment shown? This should be defined within the figure legends and it should be made clear if the error bars shown are generated from technical replicates within one experiment or from 5 independent experiments.

Reviewer 2 Report

The manuscript entitled "Ureaplasma species mitigate immune defense strategies in human brain microvasculature endothelial cells by modulation of cytokine and chemokine responses" by Silwedel et al., builds on a growing platform of research on this topic by these authors.  The number of cross-referencing methods is extensive and examines the effects of cytokine expression at both mRNA and protein levels.

There are a few limitations to the study as currently presented.  It is unclear whether viability of the Ureaplasma cultures that are altering the effects of LPS (or apoptosis or receptor expression previously published) is required or not.  The LPS will be mediating its effects via receptor binding, and while there are a range of reports showing TLR engagement by Ureaplasma, it is not clear if the effects of Ureaplasma are mediated by competing receptor engagement.  Uchida et al., 2013 showed that membrane fractions from Ureaplasma were capable of mediating foetal pathology in mice.  Given that the authors have used RNA sequencing methodology, is there any data (or possibly even a correlation) between Ureaplasma specific mRNA expression (e.g. urease genes, MBA gene or Ureaplasma house-keeping genes) are being observed between 4 and 30 hours along with the cytokine gene mRNAs?

The discrepancy between molecular and titration measurement of Ureaplasma cultures needs to be addressed.  Ureaplasma will adhere to plastic and unless the tips are changed between each well, the CCU will be artificially elevated as single colonies detach in lower dilutions.  109-1010 CCU/ml is almost impossible to achieve (especially in the absence of 10% yeast extract) without a concentration step. The molecular methods should be taken as more accurate when calculating bacterial load and MOI for the report.

The authors also cite a sheep model that first reported the synergy and antagonism of some responses to LPS when combined with Ureaplasma infection, however, the LPS was administered following chronic (70 day) or acute (7 day) infection by Ureaplasma.  It may also follow that the magnitude of the effects on the endothelial cells may be much greater if the endothelial cells were infected with Ureaplasma prior to LPS challenge, as synchronous exposure to two pathogens may not reflect what occurs in preterm birth or chorioamnionitis.  This could be discussed in the manuscript.

Inclusion of a summary table that details all of the additive, antagonistic and unchanged effects of Ureaplasma on LPS stimulation of cytokine responses would also help the reader to make sense of the large amount of data that is presented in this manuscript.

Round 2

Reviewer 1 Report

Thank you to the authors for providing detailed and thoughtful responses to each of the reviewer comments.